# Multi-Cluster Approaches to Radio Sensor Array Channel Modeling and Simulation

**DOI:** 10.3390/s24186037

**Published:** 2024-09-18

**Authors:** Xin Li, Torbjörn Ekman, Kun Yang

**Affiliations:** 1The School of Information Engineering, Zhejiang Ocean University, Zhoushan 316022, China; xin.li@zjou.edu.cn; 2The Department of Electronic Systems, Norwegian University of Science and Technology, 7491 Trondheim, Norway; torbjorn.ekman@ntnu.no

**Keywords:** AR(p) model, MIMO-OFDM, phase-shift approach, scattering cluster, spatial–temporal–spectral correlation function, state-space model

## Abstract

In this paper, we explore the physical propagation environment of radio waves by describing it in terms of distant scattering clusters. Each cluster consists of numerous scattering objects that may exhibit certain statistical properties. By utilizing geometry-based methods, we can study the channel second-order statistics (CSOS), where each distant scattering cluster corresponds to a CSOS, contributes a portion to the Doppler spectrum, and is associated with a state-space multiple-input and multiple-output (MIMO) radio channel model. Consequently, the physical propagation environment of radio waves can be modeled by summing multiple state-space MIMO radio channel models. This approach offers three key advantages: simplicity, the ability to construct the entire Doppler power spectrum from multiple uncorrelated distant scattering clusters, and the capability to obtain the channels contributed by these clusters by summing the individual channels. This methodology enables the reconstruction of the radio wave propagation environment in a simulated manner and is crucial for developing massive MIMO channel models.

## 1. Introduction

In the field of radio communications, there has been a remarkable evolution from traditional voice telephony to today’s diverse range of communication methods, including multimedia, augmented reality (AR), virtual reality (VR), mixed reality (MR), and the Internet of Everything (IoE) [1,2,3]. This historical process has highlighted a consistent trend: the growing need for higher data transfer rates, which is the fundamental driver for advances in communication technologies and methods.

One of the major challenges in radio communications is the accurate modeling of the radio channel. The radio channel consists of the effects of the electromagnetic (EM) wave propagation medium on the signal as it travels from the transmitter to the receiver. Understanding and modeling these effects is critical to the design of efficient communication systems, as they affect signal quality, data transfer rate, and overall system performance. Effective channel modeling requires consideration of various factors such as path loss, multi-path fading, shadowing, and Doppler shift, all of which can significantly alter the received signal [4,5,6].

Figure 1 depicts a typical terrestrial radio propagation environment. The basic characteristic of radio channels is fading, large-scale and small-scale fading (This refers to the concept of distance described in terms of wavelengths), which can be classified as path loss, multi-path fading, shadowing and Doppler shift on the move. Among them, multi-path fading is known as small-scale fading, and its characteristics vary with time and change over frequency and space. That is, due to the multi-path propagation of EM waves, power-limited transmitted signals will be distorted at a receiver over time, frequency, and space simultaneously [4,5,6].

To better understand the mechanisms underlying the distortion, it is essential to study this physical phenomenon in depth. In this paper, we explore the physical propagation environment of radio waves by describing it in terms of distant scattering clusters. Each cluster consists of numerous scattering objects that may exhibit certain statistical properties. By utilizing the geometry-based approach introduced in the literature [7,8,9,10], we can study the CSOS, where each distant scattering cluster corresponds to a CSOS that partially contributes to the Doppler spectrum and is associated with a state-space MIMO radio channel model. Consequently, the physical propagation environment of radio waves can be modeled by summing multiple state-space MIMO radio channel models.

The concept behind the geometry-based stochastic modeling approach involves mapping the spatial locations of scatterers within a cluster to an angular power distribution through the trigonometric relationships among the scatterers, the cluster center, and the receiver or transmitter. When the cluster follows a specific probability distribution, this angular power distribution exhibits particular statistical properties. Therefore, this method is intuitively simple and straightforward.

A distant scattering cluster leads to small variations in the angle of departure (AOD) and angle of arrival (AOA), resulting in a narrow-band Doppler spectrum at both the base station (BS) and mobile station (MS). This characteristic allows for the calculation of the CSOS using a small angular approach. This method is well-suited for decomposing the Doppler power spectrum into small, uncorrelated portions.

Figure 2 illustrates the AOA and AOD, a scattering object within a cluster, a MIMO antenna array, and the relationships among the transmitter, scatterer, and receiver, in which the BS is equipped with $M_{t}$ antenna sensors and the MS with $M_{r}$ antenna sensors. The BS remains stationary, while the MS is in motion. This diagram helps visualize the interactions and dependencies between these components in the context of radio wave propagation.

Thus, the spatial–temporal–spectral correlation function (STSCF) of a MIMO-OFDM (orthogonal frequency division multiplexing) channel can be expressed as [11,12]:(1)$$C_{h}\left(\Delta t,d_{t},d_{r},d_{f}\right)=\int_{\alpha ,\beta ,\tau }f_{\alpha ,\beta ,\tau }\left(\alpha ,\beta ,\tau \right)e^{j2\pi d_{t}\cos\left(\beta +\beta_{0}\right)}e^{j2\pi (d_{r}\cos\left(\alpha +\alpha_{0}\right)+f_{D}\Delta t\cos\left(\alpha +\alpha_{0}-\gamma \right))}e^{-j2\pi d_{f}\tau }d\alpha d\beta d\tau$$
where $f_{D}$ is the Doppler frequency, Δt is the time separation, and $f_{D}\Delta t$ represents the MS displacement. Angles β and α signify the AOD and AOA, respectively, with means denoted by $\beta_{0}$ and $\alpha_{0}$. The angle γ is between the moving direction and the antenna array, as illustrated in Figure 2. At the BS, $d_{t}=m_{t}\Delta d_{t}$ denotes antenna spacing, where $m_{t}\in \left\{0,1,\cdots ,M_{t}-1\right\}$ and $\Delta d_{t}$ is the spacing between two adjacent antenna sensors, and at the MS, $d_{r}=m_{r}\Delta d_{r}$ represents the antenna spacing, where $m_{r}\in \left\{0,1,\cdots ,M_{r}-1\right\}$, and $\Delta d_{r}$ is the spacing between two adjacent antenna sensors. The frequency separation $d_{f}=m_{f}\Delta f$, with sub-carrier frequencies defined as $f_{i}=f_{c}+i\Delta f$, in which $m_{f}\in \left\{0,1,2,\cdots ,M_{f}-1\right\}$, and $f_{c}$ is the carrier frequency. In the case of $m_{f}=0$, it signifies a single-carrier modulation system, a MIMO system. The joint angular-delay power distribution function is denoted as $f_{\alpha ,\beta ,\tau }\left(\alpha ,\beta ,\tau \right)$, where τ represents the time delay.

Assuming independence among α, β, and τ, the joint angular-delay power distribution function $f_{\alpha ,\beta ,\tau }\left(\alpha ,\beta ,\tau \right)$ can be factorized as $f_{\alpha }\left(\alpha \right)f_{\beta }\left(\beta \right)f_{\tau }\left(\tau \right)$. Here, $f_{\alpha }\left(\alpha \right)$ represents the angular power distribution function (APDF) of the AOA, $f_{\beta }\left(\beta \right)$ denotes the APDF of the AOD, and $f_{\tau }\left(\tau \right)$ is the delay power distribution function (DPDF) corresponding to the time of arrival (TOA).

This assumption is justified as radio signals typically pass through multiple scatterers within a cluster when propagating from a transmitter to a receiver, implying independence between α and β. Obviously, τ is inherently independent of both α and β.

Using Taylor expansion for small angles, Equation (Equation 1) is simplified to:(2)$$\begin{array}{cc} & C_{h}\left(\Delta t,d_{t},d_{r},d_{f}\right)\approx {\bar{C}}_{h}\left(\Delta t,d_{t},d_{r},d_{f}\right)=\int_{\alpha ,\beta ,\tau }f_{\alpha }\left(\alpha \right)f_{\beta }\left(\beta \right)f_{\tau }\left(\tau \right)e^{j2\pi d_{t}\cos\left(\beta_{0}\right)}\\ \; & \;\;\;\;\;\;\times e^{-j2\pi d_{t}\;\sin\left(\beta_{0}\right)\beta }e^{j2\pi (d_{r}\;\cos\left(\alpha_{0}\right)+f_{D}\Delta t\;\cos\left(\alpha_{0}-\gamma \right))}e^{-j2\pi (d_{r}\;\sin\left(\alpha_{0}\right)+f_{D}\Delta t\;\sin\left(\alpha_{0}-\gamma \right))\alpha }e^{-j2\pi d_{f}\tau }d\alpha d\beta d\tau \end{array}$$
with special cases for the channel temporal dynamic function ${\bar{R}}_{h}\left(\Delta t\right)$:(3)$${\bar{R}}_{h}\left(\Delta t\right)={\bar{C}}_{h}\left(\Delta t,0,0,0\right)=\int_{\alpha }f_{\alpha }\left(\alpha \right)e^{j2\pi f_{D}\Delta t\cos\left(\alpha_{0}-\gamma \right)}e^{-j2\pi f_{D}\Delta t\sin\left(\alpha_{0}-\gamma \right)\alpha }d\alpha$$
and the spatial–temporal correlation function (STCF) ${\bar{C}}_{h}\left(\Delta t,d_{t},d_{r},0\right)$:(4)$$\begin{array}{cc} & {\bar{C}}_{h}\left(\Delta t,d_{t},d_{r},0\right)=\int_{\beta }f_{\beta }\left(\beta \right)e^{j2\pi d_{t}\cos\left(\beta_{0}\right)}e^{-j2\pi d_{t}\sin\left(\beta_{0}\right)\beta }d\beta \\ \; & \;\;\;\;\;\;\times \int_{\alpha }f_{\alpha }\left(\alpha \right)e^{j2\pi (d_{r}\cos\left(\alpha_{0}\right)+f_{D}\Delta t\cos\left(\alpha_{0}-\gamma \right))}e^{-j2\pi (d_{r}\sin\left(\alpha_{0}\right)+f_{D}\Delta t\sin\left(\alpha_{0}-\gamma \right))\alpha }d\alpha \end{array}$$

Radio wave measurements indicate that the power azimuth spectrum typically exhibits sharp, narrow peaks over a small range of angles [9,10,13,14]. Each has been modeled as a Laplace angular distribution [9,10,15,16,17]. Some measurements display smooth peaks [13,18], which could be modeled by other distributions.

In this paper, the sharp peaks in the angular spectrum are modeled as Cauchy APDF and the smooth peaks are modeled as Gaussian APDF. These distributions allow for integrable expressions of the STCF, Equation (Equation 4). Other distributions can be approximated as weighted sums of Gaussian APDFs or the combination of Gaussian and Cauchy APDFs [19].

Although the Cauchy probability density function (PDF) exhibits heavier tails in comparison to the Laplace PDF, as shown in Figure 3a,b, it is still a viable option if the majority of power (e.g., 90% or more) is concentrated within a narrower angular range. In a geometric context, this can be conceptualized as the main scattering objects being situated near the center of a cluster, while the remaining objects contribute a significantly lesser amount of power to the antennas and are therefore worth ignoring. This concept is equally applicable to truncated Gaussian PDFs.

It has been identified that, for a Cauchy APDF, the corresponding clusters have the following property: the distance between the cluster center and the scattering objects obeys the Cauchy–Rayleigh distribution [20], and for a Gaussian APDF, the corresponding clusters have the property that the distance between the cluster center and the scatterers follows the Rayleigh distribution [21]. They are named the Cauchy–Rayleigh cluster and Rayleigh cluster, respectively, and their properties can be used for MIMO channel characterization.

Based on the trigonometric relationships between the transmitter, scatterers, and receiver, the APDFs of these two types of scattering clusters can be derived. Additionally, the spatial–temporal correlation function is integrable according to the obtained APDF. The analytical solution, or closed-form solution, will be associated with a distant scattering cluster, meaning that the solution will depend on the characteristics of a given geometrical cluster.

Furthermore, to characterize the MIMO channel using a state-space representation, the correlation function needs to be separated into two disjoint parts: a temporal term that accounts for the movement and a spatial term that covers the antenna configuration. This separation facilitates a more accurate and manageable representation of the channel dynamics, capturing the essential features of the spatial and temporal variations in the propagation environment.

However, the analytical solution based on the Cauchy APDF involves an absolute sum of terms in the exponent associated with antenna spacing and movement. It requires removing the sign from the absolute value. Conversely, the solution based on the Gaussian APDF introduces a cross-term related to antenna spacing and motion. This cross-term does not fall distinctly into the categories of channel temporal dynamics or spatial correlation [20,21].

Unlike work found in the literature [20,21] where antenna spacing and movement are separated, in this paper, to effectively separate antenna spacing and movement, we propose a linear transformation, termed the phase-shift method. This method avoids unnecessary approximations and provides a clean separation, making the CSOS expression integrable.

The elegance of this approach is that it makes the CSOS expression integrable. The analytical solution can be decomposed into two disjoint parts: the temporal dynamics and spatial correlation of the channel. The temporal dynamics can be modeled using autoregressive models, while the spatial correlation is characterized using the Kronecker matrix.

Depending on the cluster type, the temporal dynamics can be appropriately modeled as a first-order autoregressive model (AR(1)) or an AR(3) model. The AR(2) model can also be used if the approximation requirements are considered acceptable. Thus, this approach is useful for constructing state-space models of MIMO/massive MIMO channels [22,23,24], as well as multi-cluster state-space MIMO-OFDM channel models [25].

This paper explores a cluster-based propagation environment and validates the theoretical model through simulations. This approach consists of four main steps:Derivation of the mathematical representation of the phase-shift CSOSWe derive the mathematical expression for the phase-shift CSOS, which captures the channel properties within a different system in the presence of scattering clusters.Development of MIMO radio channel models in state-space formWe develop MIMO radio channel models using state-space representations. This involves formulating the system dynamics and observation equations to model the radio channel behavior.Reconstruction of the physical propagation environmentWe reconstruct the physical propagation environment of radio waves by applying multiple MIMO-OFDM channel models. This step integrates the contributions from various clusters to simulate a realistic propagation scenario.SimulationsFinally, we validate the theoretical model through extensive simulations. This step involves generating synthetic data based on our model and calculating the CSOS. We then compare the calculated CSOS values with the theoretical predictions to ensure the accuracy and reliability of the model.

Although the results presented in this paper are based on two dimensions, this is the first step in our radio channel modeling. By introducing another dimension, as described in the literature [26], this approach can be more easily extended to three dimensions.

The rest of the paper is organized as follows: Section 2 introduces the theoretical framework. Section 3 presents the simulation results. Finally, the conclusions are provided in Section 4.

## 2. Theoretical Framework

In this section, we outline the basic concepts and theoretical models that underpin our simulations. We first derive a mathematical representation of the phase-shift CSOS, which plays a key role in channel modeling. This is followed by a geometric-based approach to obtain an analytical solution of the CSOS by mapping scattering objects within a cluster into the designed AOA/AOD from a spatial perspective. Finally, we will delve into the multi-cluster MIMO-OFDM channel model. The model is particularly well suited to capture the intricacies of wireless communications in multi-cluster scenarios. Together, these elements offer a comprehensive theoretical foundation for simulating a radio wave propagation environment.

### 2.1. Linear Transformation

From a mathematical standpoint, our approach involves transforming the current Cartesian coordinate system into an alternative one. In this new system, antenna spacing and movement components are separated into disjoint, error-free parts, allowing for the modeling of channel characteristics through a state-space representation. An inverse linear mapping is then applied to revert the channel properties to their original form within the original coordinate system.

Equation (Equation 4) approximates the correlation related to MS antenna sensors by considering the AOA near zero degrees around the angles $\alpha_{0}$ and $\alpha_{0}-\gamma$. This approximation, illustrated in Figure 4, decomposes the movement and antenna spacing into a phase change on OA and a damping change on AW, with OA and AW orthogonal.

Given $\mathrm{AG}=d_{\mathrm{AG}}$, $\mathrm{GH}=d_{\mathrm{GH}}$, $\mathrm{AU}=d_{\mathrm{AU}}$ and $\mathrm{UW}=d_{\mathrm{UW}}$, we obtain:(5)$$\begin{array}{cc} d_{\mathrm{AG}}+d_{\mathrm{GH}} & =s\cos\left(\alpha_{0}-\gamma \right)+d_{r}\cos\left(\alpha_{0}\right)\\ d_{\mathrm{AU}}+d_{\mathrm{UW}} & =s\sin\left(\alpha_{0}-\gamma \right)+d_{r}\sin\left(\alpha_{0}\right) \end{array}$$

Geometrically, these equations interpret the meaning of the approximation in Equation (Equation 4).

Alternatively, AW can be viewed as the projection of AD, with $\mathrm{AD}=d_{\mathrm{AD}}=\kappa$. Using the right triangle relationship, we obtain the following:(6)$$\kappa =\frac{s\sin\left(\alpha_{0}-\gamma \right)+d_{r}\sin\left(\alpha_{0}\right)}{\cos(90{}^{\circ}-\alpha_{0}+\gamma )}=s+d_{r}\frac{\sin(\alpha_{0})}{\sin(\alpha_{0}-\gamma )}$$

This implies that while antenna sensor A moves to D, its actual position is at E, determining the changed phase with the following:(7)$$\begin{array}{cc} d_{\mathrm{AC}}+d_{\mathrm{GH}}-d_{\mathrm{GC}} & =\kappa \cos\left(\alpha_{0}-\gamma \right)+d_{r}\cos\left(\alpha_{0}\right)-d_{r}\frac{\sin(\alpha_{0})}{\sin(\alpha_{0}-\gamma )}\cos\left(\alpha_{0}-\gamma \right)\\ \; & =\kappa \cos\left(\alpha_{0}-\gamma \right)-d_{r}\frac{\sin(\gamma )}{\sin(\alpha_{0}-\gamma )} \end{array}$$

Notably, $\alpha_{0}-\gamma \neq 0$ or π, as the transformation in Equation (Equation 6) would otherwise be meaningless.

In the new system, Equation (Equation 6), the antenna spacing becomes part of the motion. Assuming $\kappa =f_{D}\Delta t_{\kappa }$, Equation (Equation 6) becomes as follows:(8)$$\Delta t_{\kappa }-\Delta t=d_{r}\frac{\sin(\alpha_{0})}{f_{D}\sin\left(\alpha_{0}-\gamma \right)}$$

From a delay perspective, as shown in Equation (Equation 8), the time difference between these systems implies that antenna spacing becomes a time delay in channel dynamics, linking the new system to the original system.

Substituting Equation (Equation 8) into Equation (Equation 4), we obtain the following:(9)$$\begin{array}{cc} {\bar{C}}_{h}^{\kappa }\left(\Delta t_{\kappa },d_{t},d_{r},0\right)= & \;e^{-j2\pi d_{r}\sin\left(\gamma \right)/\sin\left(\alpha_{0}-\gamma \right)}\int_{\beta }f_{\beta }\left(\beta \right)e^{-j2\pi d_{t}\sin\left(\beta_{0}\right)\beta }d\beta \\ \; & \;\times e^{j2\pi d_{t}\cos\left(\beta_{0}\right)}e^{j2\pi f_{D}\Delta t_{\kappa }\cos\left(\alpha_{0}-\gamma \right)}\int_{\alpha }f_{\alpha }\left(\alpha \right)e^{-j2\pi f_{D}\Delta t_{\kappa }\sin\left(\alpha_{0}-\gamma \right)\alpha }d\alpha \end{array}$$

In this new system, the spatial correlation of MS-associated channels is depicted through a phase rotation, effectively separating movement and antenna spacing into distinct entities. This phase rotation remains independent of the BS antenna, allowing for the use of the Kronecker product to construct state-space MIMO channels [27].

The phase-shift method offers an alternative perspective on the problem. By changing variables, a straightforward, error-free method emerges for separating movement and antenna spacing. The channel STCF can thus be conceptualized as the outcome of the phase rotation multiplied by the channel’s temporal dynamics, offset by a specific value along the direction of motion.

### 2.2. Geometry Based Approach

Research on a geometry-based MIMO radio channel modeling has garnered significant attention [7,8,9,10,28]. This approach involves mapping the spatial locations of scatterers within a cluster to an angular power distribution using trigonometric relationships among the scatterers, the cluster center, and the receiver or transmitter. Additionally, the angular power distribution exhibits specific statistical properties when the cluster follows a defined probability distribution [7,8]. This method is both simple and intuitive.

In this subsection, we will derive the analytical solutions for the CSOS in the new coordinate system, providing insights into the spatial correlation and channel characteristics of MIMO systems in various propagation environments.

#### 2.2.1. Cauchy–Rayleigh Cluster

For a distant Cauchy–Rayleigh cluster, the Cauchy APDFs can be derived from the geometric relations, as shown in Figure 2 [25]:(10)$$f_{\alpha }\left(\alpha \right)\approx f_{\alpha }^{c}\left(\alpha \right)=\frac{1}{\pi }\frac{\eta_{r}}{\eta_{r}^{2}+\alpha^{2}},\;\;\;\;\;f_{\beta }\left(\beta \right)\approx f_{\beta }^{c}\left(\beta \right)=\frac{1}{\pi }\frac{\eta_{t}}{\eta_{t}^{2}+\beta^{2}}$$
where ${\left[\cdot \right]}^{c}$ indicates that the APDF is obtained in terms of the Cauchy–Rayleigh cluster, the parameters $\eta_{t}=\zeta /d_{{\mathrm{OB}}_{1}}$ and $\eta_{r}=\zeta /d_{{\mathrm{OM}}_{1}}$, $d_{{\mathrm{OB}}_{1}}={\mathrm{OB}}_{1}$ and $d_{{\mathrm{OM}}_{1}}={\mathrm{OM}}_{1}$ are used to control the angular width of these two distributions, respectively. ζ>0 is the dispersion of the Cauchy–Rayleigh distribution.

Obviously, both $f_{\alpha }^{c}\left(\alpha \right)$ and $f_{\beta }^{c}\left(\beta \right)$ in Equation (Equation 10) are defined on [−π,π], which means that they are not proper Cauchy angular power density functions because the Cauchy probability density function is defined over the infinite interval. To integrate these functions effectively, we extend the integral from [−π,π] to (−∞,∞), as described in detail in the book [25].

Therefore, the STCF can be given by the following:(11)$${\tilde{C}}_{h}^{c_{\kappa }}\left(\Delta t_{\kappa },d_{t},d_{r},0\right)=e^{-j2\pi d_{r}\sin\left(\gamma \right)/\sin\left(\alpha_{0}-\gamma \right)}e^{-2\pi \eta_{t}d_{t}\left|\sin\left(\beta_{0}\right)\right|}e^{j2\pi d_{t}\cos\left(\beta_{0}\right)}e^{-2\pi \eta_{r}f_{D}\Delta t_{\kappa }\left|\sin\left(\alpha_{0}-\gamma \right)\right|}e^{j2\pi f_{D}\Delta t_{\kappa }\cos\left(\alpha_{0}-\gamma \right)}$$

The channel dynamic function is as follows:(12)$${\tilde{R}}_{h}^{c_{\kappa }}\left(\Delta t_{\kappa }\right)={\tilde{C}}_{h}^{c_{\kappa }}\left(\Delta t_{\kappa },0,0,0\right)=e^{-2\pi \eta_{r}f_{D}\Delta t_{\kappa }\left|\sin\left(\alpha_{0}-\gamma \right)\right|}e^{j2\pi f_{D}\Delta t_{\kappa }\cos\left(\alpha_{0}-\gamma \right)}$$

According to Equation (Equation 11), we can assign a specific expression to each element of $R_{\mathrm{BS}}$ in [29,30]:(13)$$r_{m,n}^{c,\mathrm{BS}}\left(d_{t}\right)=e^{-2\pi \eta_{t}d_{t}\left|\sin\left(\beta_{0}\right)\right|}e^{j2\pi d_{t}\cos\left(\beta_{0}\right)}$$
where $r_{m,n}^{\mathrm{BS}}$ replaces the notation $\rho_{m_{1},m_{2}}^{\mathrm{BS}}$ in [29], and the notation $R_{\mathrm{BS}}^{c}$ also replaces $R_{\mathrm{BS}}$. Similarly, all elements of $R_{\mathrm{MS}}$ in [29] will have the following specific expression:(14)$$r_{i,j}^{\mathrm{MS}}\left(d_{r}\right)=e^{-j2\pi d_{r}\sin\left(\gamma \right)/\sin\left(\alpha_{0}-\gamma \right)}$$

This shows that the spatial correlation between MS channels depends only on antenna spacing $d_{r}$ but not on the cluster type, maintaining the notation $R_{\mathrm{MS}}$.

#### 2.2.2. Rayleigh Cluster

Similarly, for a distant Rayleigh cluster, we obtain the following approximate Gaussian APDFs [25]:(15)$$f_{\alpha }^{r}\left(\alpha \right)=\frac{1}{\sqrt{2\pi }\sigma_{r}}e^{-\frac{\alpha^{2}}{2\sigma_{r}^{2}}},\;\;\;\;\;f_{\beta }^{r}\left(\beta \right)=\frac{1}{\sqrt{2\pi }\sigma_{t}}e^{-\frac{\beta^{2}}{2\sigma_{t}^{2}}}$$
where ${\left[\cdot \right]}^{r}$ indicates that the APDF is obtained from the Rayleigh cluster, $\sigma_{t}=\sigma /d_{{\mathrm{OB}}_{1}}$, $\sigma_{r}=\sigma /d_{{\mathrm{OM}}_{1}}$, and σ is from the Rayleigh distribution.

These truncated Gaussian APDFs can be extended from π to infinity, and the analytical solution of the CSOS is obtained by substituting Equation (Equation 15) into Equation (Equation 9):(16)$${\tilde{C}}_{h}^{r_{\kappa }}\left(\Delta t_{\kappa },d_{t},d_{r},0\right)=e^{-j2\pi d_{r}\sin\left(\gamma \right)/\sin\left(\alpha_{0}-\gamma \right)}e^{-2\pi^{2}\sigma_{t}^{2}d_{t}^{2}\sin^{2}\left(\beta_{0}\right)}e^{j2\pi d_{t}\cos\left(\beta_{0}\right)}e^{-2\pi^{2}\sigma_{r}^{2}\sin^{2}\left(\alpha_{0}-\gamma \right)f_{D}^{2}\Delta t_{\kappa }^{2}}e^{j2\pi \cos\left(\alpha_{0}-\gamma \right)f_{D}\Delta t_{\kappa }}$$

Thus, the channel temporal dynamic function is as follows:(17)$${\tilde{R}}_{h}^{r_{\kappa }}\left(\Delta t_{\kappa }\right)=e^{-2\pi^{2}\sigma_{r}^{2}\sin^{2}\left(\alpha_{0}-\gamma \right)f_{D}^{2}\Delta t_{\kappa }^{2}}e^{j2\pi \cos\left(\alpha_{0}-\gamma \right)f_{D}\Delta t_{\kappa }}$$
each element of $R_{\mathrm{BS}}^{r}$ can be given by the following:(18)$$r_{m,n}^{r,\mathrm{BS}}\left(d_{t}\right)=e^{-2\pi^{2}\sigma_{t}^{2}d_{t}^{2}\sin^{2}\left(\beta_{0}\right)}e^{j2\pi d_{t}\cos\left(\beta_{0}\right)}$$
and all elements of $R_{\mathrm{MS}}^{r}$ are also given by Equation (Equation 14).

### 2.3. Multi-Cluster MIMO-OFDM Channel Model

In this subsection, we will examine channel characterization in a multi-cluster environment. The typical physical propagation environment of radio waves is illustrated in Figure 1 and can be modeled as depicted in Figure 5.

Consider a radio wave propagation scenario characterized by *K* distant scattering clusters, as shown in Figure 5. These clusters are modeled using a combination of Cauchy–Rayleigh and Rayleigh distributions. The BS remains stationary while the MS moves at a speed *v*. There is no line-of-sight (NLOS) between the BS and the MS, and all transmitted and received signals occur through these *K* uncorrelated scattering clusters. Each cluster is decomposed into a number of resolvable multi-path components [25], and the trigonometric relationships among the BS, scatterers, and MS within a cluster are illustrated in Figure 2.

In this model, the radio waves from different clusters can be summed to account for the contributions from all clusters. The power emitted from each cluster is apportioned to the Doppler power spectrum, making the combination of radio waves resemble the sum of individual power contributions. This yields a *K*-cluster MIMO channel model. Furthermore, when the delay factor is considered, these contributions lead to a *K*-cluster MIMO-OFDM channel model.

#### 2.3.1. Multi-Cluster Angular-Delay Spectrum

The joint angular-delay spectrum associated with *K* scattering clusters can be expressed as follows:(19)$$f_{\alpha ,\tau }\left(\alpha ,\tau \right)=\frac{\sum_{k=1}^{K}P_{k}f_{\alpha_{k},\tau_{k}}\left(\alpha_{k},\tau_{k}\right)}{\sum_{k=1}^{K}P_{k}}=\frac{\sum_{k=1}^{K}P_{k}f_{\alpha_{k}}\left(\alpha_{k}\right)f_{\tau_{k}}\left(\tau_{k}\right)}{\sum_{k=1}^{K}P_{k}}$$
where $P_{k}$ denotes the power contributed from the *k*th cluster. Summing over the angles, the marginal distribution represents the power delay profile (PDP), $f_{\tau }\left(\tau \right)$, of the clusters. Summing over the delays yields the angular power distribution, $f_{\alpha }\left(\alpha \right)$, of the clusters [25].

Next, we will introduce the step-by-step construction of a multi-cluster MIMO-OFDM channel model starting from the AR(p) model.

#### 2.3.2. AR(p) Model

As presented in Section 2.2, a channel STCF is obtained based on the distant scattering clusters, as shown in Equation (Equation 11) or Equation (Equation 16). Here, its dynamic part will be modeled using the following AR(p) model [31],
(20)$$x_{k}=\sum_{i=1}^{p}\phi_{i}x_{k-i}+w_{k}$$
where $\phi_{1},\phi_{2},\cdots ,\phi_{p}$ ($\phi_{p}\neq 0$) are complex coefficients (weights), and the innovation noise $w_{k}$ is a complex Gaussian sequence $\mathrm{CN}(0;\sigma_{w}^{2})$. That is, the stochastic variable $x_{k}$ is defined as a linear combination of its previous p values of the series plus an innovation noise.

The channel dynamics associated with the Cauchy–Rayleigh clusters are modeled as an AR(1) model [19,20], while the channel dynamics associated with the Rayleigh clusters can be approximated as an AR(3) model [21]. The coefficients $\phi_{i}$ for these models can be estimated using the least-squares (LS) method [32] or computed using the spectral-equivalent (SE) method [21].

Thus, a single peak on the Doppler spectrum corresponding to the contribution from a distant scattering cluster is modeled by an AR(p) model. The advantage of using an AR(p) model is that it can be directly parameterized by the properties of the cluster and allows changing the angles in the simulation, which corresponds to changing the directions of the mobile receiver.

#### 2.3.3. SISO Channel Model

An AR(p) model is described by the block diagram shown in Figure 6 and can be implemented in the controlled canonical representation to form a state-space single-input and single-output (SISO) channel model [33]:(21)$$\begin{array}{cc} x_{k+1} & =Ax_{k}+Bw_{k}\\ h_{k} & =Cx_{k} \end{array}$$
where C=[00⋯01] is a 1×p vector, $B={\left[0\;0\;\cdots \;0\;1\right]}^{T}$ is a p×1 vector, the output $h_{k}=x_{k}$ is a scalar, the channel, and
(22)$$\begin{array}{cc} x_{k+1}=\left[\begin{array}{c} x_{k-p+1}\\ x_{k-p+2}\\ \vdots \\ x_{k}\\ x_{k+1} \end{array}\right],\;\; & A=\left[\begin{array}{ccccc} 0 & 1 & 0 & \cdots & 0\\ 0 & 0 & 1 & \cdots & 0\\ \vdots & \vdots & \vdots & \ddots & \vdots \\ 0 & 0 & 0 & \cdots & 1\\ \phi_{p} & \phi_{p-1} & \phi_{p-2} & \cdots & \phi_{1} \end{array}\right],\\ \; & x_{k}=\left[\begin{array}{c} x_{k-p}\\ x_{k-p+1}\\ \vdots \\ x_{k-1}\\ x_{k} \end{array}\right] \end{array}$$

Other assumptions include: $w_{k}∼\mathrm{CN}\left(0;\sigma_{w}^{2}\right)$, where $\sigma_{w}^{2}=1$, and the initial state $x_{0}$ is uncorrelated to the scalar complex Gaussian noise $w_{k}$.

The input vector $B={\left[0\;0\;\cdots \;0\;\sigma_{\mathrm{AR}(p)}\right]}^{T}$ is redefined so that the noise input will be scaled by $\sigma_{\mathrm{AR}(p)}$. This adjustment ensures that the variance of $x_{k}$ is unity, i.e., $\sigma_{x}^{2}=1$. It is logical to have unit variance before C and allow C to scale the contribution from a cluster, incorporating path loss to $h_{k}$ [25].

It is important to note that, in practice, the matrices A and B will exhibit time-varying characteristics. This time variation stems from the angle-related elements inherent in these two matrices. The angle $\alpha_{0}-\gamma$ is used to describe the direction of dynamic motion towards the center of the cluster, which changes continuously throughout the motion.

In this paper, an assumption is made that these matrices remain time-invariant due to the insignificance of the movements compared to the distance between the MS and the center of a scattering cluster. Essentially, within certain time slots, all matrices can be considered approximately constant. This assumption implies constant angles towards clusters, constant speed during movement, and thus a time-invariant environment is achieved. This assumption is related to the stationarity of $x_{k}$ and $h_{k}$ sequences as well.

#### 2.3.4. MIMO Channel Model

Based on the SISO channel model block depicted in Figure 6, a state-space SIMO channel model is constructed by connecting multiple SISO channel model blocks in parallel. A correlated innovation process is employed to adjust the spatial correlation between these SISO channel blocks. The number of SISO channel model blocks required for the SIMO channel model depends on the number of receiving antenna elements $M_{r}$. An example of constructing a 1×2 SIMO channel model is provided in [20].

Similarly, by connecting multiple SIMO channel blocks in parallel, a state-space MIMO channel model is constructed, as shown in Figure 7. Here, $\Phi_{M_{r}M_{t}}$ denotes the coloring matrix. The number of SIMO channel blocks needed for the MIMO channel model depends on the number of transmitting antennas $M_{t}$.

Mathematically, this block diagram can be implemented as the following state-space representation [25]:(23)$$\begin{array}{cc} x_{k+1} & =\Gamma^{mimo}x_{k}+\Psi^{mimo}w_{k}\\ h_{k}^{mimo} & =\Omega^{mimo}x_{k} \end{array}$$
where $x_{k}\in C^{{\mathrm{pM}}_{r}M_{t}}$, $w_{k}∼\mathrm{CN}\left(0;1\right)\in C^{M_{r}M_{t}}$ and $h_{k}^{mimo}=\mathrm{vec}\left(H_{k}\right)$, here
(24)$$H_{k}=\left[\begin{array}{cccc} h_{k}\left[1,1\right] & h_{k}\left[1,2\right] & \cdots & h_{k}\left[1,M_{t}\right]\\ h_{k}\left[2,1\right] & h_{k}\left[2,2\right] & \cdots & h_{k}\left[2,M_{t}\right]\\ \vdots & \vdots & \ddots & \vdots \\ h_{k}\left[M_{r},1\right] & h_{k}\left[M_{r},2\right] & \cdots & h_{k}\left[M_{r},M_{t}\right] \end{array}\right]$$The matrices $\Gamma^{mimo}$, $\Psi^{mimo}$ and $\Omega^{mimo}$ are defined by the following:(25)$$\begin{array}{cc} \Gamma^{mimo} & =I_{M_{r}M_{t}}⊗A\\ & ={\left[\begin{array}{cccc} A & 0 & \cdots & 0\\ 0 & A & \cdots & 0\\ \vdots & \vdots & \ddots & \vdots \\ 0 & 0 & \cdots & A \end{array}\right]}_{{\mathrm{pM}}_{r}M_{t}\times {\mathrm{pM}}_{r}M_{t}}\\ \Psi^{mimo} & =\left(I_{M_{r}M_{t}}⊗B\right)\Phi_{M_{r}M_{t}}\\ & ={\left[\begin{array}{cccc} B & 0 & \cdots & 0\\ 0 & B & \cdots & 0\\ \vdots & \vdots & \ddots & \vdots \\ 0 & 0 & \cdots & B \end{array}\right]}_{{\mathrm{pM}}_{r}M_{t}\times M_{r}M_{t}}\Phi_{M_{r}M_{t}}\\ \Omega^{mimo} & =I_{M_{r}M_{t}}⊗C\\ \; & ={\left[\begin{array}{cccc} C & 0 & \cdots & 0\\ 0 & C & \cdots & 0\\ \vdots & \vdots & \ddots & \vdots \\ 0 & 0 & \cdots & C \end{array}\right]}_{M_{r}M_{t}\times {\mathrm{pM}}_{r}M_{t}} \end{array}$$
where $I_{M_{r}M_{t}}$ denotes the identity matrix of size $M_{r}M_{t}$, ⊗ denotes the Kronecker operator, and $\Phi_{M_{r}M_{t}}$, employed to control the number of $M_{r}M_{t}$ correlated driving noises, is defined as a lower triangular matrix:(26)$$\begin{array}{c} \Phi_{M_{r}M_{t}}=\left[\begin{array}{cccc} \varphi_{1,1} & 0 & \cdots & 0\\ \varphi_{2,1} & \varphi_{2,2} & \cdots & 0\\ \vdots & \vdots & \ddots & \vdots \\ \varphi_{M_{r}M_{t},1} & \varphi_{M_{r}M_{t},2} & \cdots & \varphi_{M_{r}M_{t},M_{r}M_{t}} \end{array}\right] \end{array}$$

A straightforward computation leads to the following equation:(27)$$\begin{array}{cc} R_{h} & =E\left[h_{k}^{mimo}h_{k}^{mimo^{H}}\right]=\Phi_{M_{r}M_{t}}\Phi_{M_{r}M_{t}}^{H}\\ \; & =R_{\mathrm{MIMO}}=R_{\mathrm{BS}}⊗R_{\mathrm{MS}} \end{array}$$

The Cholesky decomposition method can be used to solve Equation (Equation 27) numerically. However, for a small size matrix $\Phi_{M_{r}M_{t}}$, like a 2×2 MIMO channel model, an analytical solution of a lower triangular matrix $\Phi_{4}$ is obtained [21].

#### 2.3.5. MIMO-OFDM Channel Model

The MIMO channel model described in Section 2.3.4 is applicable to narrow-band and single-carrier frequency scenarios. In this section, we propose a viable approach to address the issue of frequency-selective fading channels. Specifically, we introduce a wide-band MIMO channel model, known as the MIMO-OFDM channel model, which integrates MIMO and OFDM techniques.

To achieve this, we incorporate a parameter called the delay factor to account for the delay spread resulting from two-dimensional (2D) scattering clusters. This modification aims to highlight the spatial–temporal–spectral correlation of the channel, taking into account not only the spatial–temporal correlation but also the spectral characteristics.

Utilizing the MIMO channel model framework illustrated in Figure 7, we construct a MIMO-OFDM channel model as outlined in Figure 8. In this scenario, we introduce a colored input noise vector for the MIMO-OFDM channels. This noise vector is generated using the spectral correlation matrix $C_{f}$, defined as follows [25]:(28)$$C_{f}=\left[\begin{array}{cccc} r_{f}\left[0\right] & r_{f}\left[1\right] & \cdots & r_{f}\left[M_{f}-1\right]\\ r_{f}^{*}\left[1\right] & r_{f}\left[0\right] & \cdots & r_{f}\left[M_{f}-2\right]\\ \vdots & \vdots & \ddots & \vdots \\ r_{f}^{*}\left[M_{f}-1\right] & r_{f}^{*}\left[M_{f}-2\right] & \cdots & r_{f}\left[0\right] \end{array}\right]$$
where each element is calculated by
(29)$$r_{f}\left(d_{f}\right)=\int_{\tau }f_{\tau }\left(\tau \right)e^{-j2\pi d_{f}\tau }d\tau$$
which is denoted the discrete form by $r_{f}\left[m_{f}\right]$. Clearly, the diagonal element $r_{f}\left[0\right]=\int_{\tau }f_{\tau }\left(\tau \right)d\tau =1$.

Next, we will first show how to compute $r_{f}\left[m_{f}\right]$ in Equation (Equation 29) and then present the construction of a state-space MIMO-OFDM channel model.

Given the Cauchy–Rayleigh clusters, the corresponding Cauchy delay power distribution function of the TOA is as follows [25]:(30)$$f_{\tau }^{c}\left(\tau \right)\approx f_{\tau }\left(\tau \right)=\frac{1}{\pi }\frac{\eta }{\eta^{2}+{\left(\tau -\bar{\tau }\right)}^{2}}$$
where $\bar{\tau }$ denotes the average time delay:(31)$$\begin{array}{cc} \bar{\tau } & =\frac{d_{{\mathrm{OB}}_{1}}+d_{{\mathrm{OM}}_{1}}}{v_{c}},\;\;\;\eta =\frac{\zeta \sqrt{2+2\cos(\theta_{0})}}{v_{c}}\\ \cos(\theta_{0}) & =\frac{d_{{\mathrm{OB}}_{1}}^{2}+d_{{\mathrm{OM}}_{1}}^{2}-d_{B_{1}M_{1}}^{2}}{2d_{{\mathrm{OB}}_{1}}d_{{\mathrm{OM}}_{1}}} \end{array}$$$v_{c}$ denotes the speed of light, $\theta_{0}$ is the angle between the two edges $d_{{\mathrm{OB}}_{1}}$ and $d_{{\mathrm{OM}}_{1}}$, as illustrated in Figure 2. In Section 2.2.1, we obtained $\zeta =\alpha_{90\%}d_{{\mathrm{OM}}_{1}}/6.3138$. Hence, the following is obtained:(32)$$\eta =\frac{\alpha_{90\%}\sqrt{2+2\cos(\theta_{0})}}{6.3138}\frac{d_{{\mathrm{OM}}_{1}}}{v_{c}}$$

Substituting Equation (Equation 30) into Equation (Equation 29) gives the following:(33)$$r_{f}^{c}\left(d_{f}\right)=e^{-2\pi \eta d_{f}}e^{-j2\pi d_{f}\bar{\tau }}$$

Given a distant Rayleigh cluster, the approximate Gaussian delay power distribution function of the TOA is obtained [25]:(34)$$f_{\tau }^{r}\left(\tau \right)\approx f_{\tau }\left(\tau \right)=\frac{1}{\sqrt{2\pi }\sigma_{0}}e^{-\frac{{\left(\tau -\bar{\tau }\right)}^{2}}{2\sigma_{0}^{2}}}$$
where
(35)$$\sigma_{0}=\frac{\sigma \sqrt{2+2\cos(\theta_{0})}}{v_{c}}$$
and $\cos(\theta_{0})$, $\bar{\tau }$ are defined in Equation (Equation 31). Therefore, for Gaussian distributed TOA, we have
(36)$$r_{f}^{r}\left(d_{f}\right)=e^{-2\pi^{2}\sigma_{0}^{2}d_{f}^{2}}e^{-j2\pi \bar{\tau }d_{f}}$$

The vector ${\left[h{\left(t,f_{0}\right)}^{T}\;h{\left(t,f_{1}\right)}^{T}\;\cdots \;h{\left(t,f_{M_{f}-1}\right)}^{T}\right]}^{T}$ is used to represent all of the MIMO-OFDM channels. This formulation results in the channels being characterized by a STSCF. This STSCF captures the dependencies and variations in the channel across time, space, and frequency, providing a comprehensive model of the MIMO-OFDM channels.

In Figure 8, h[k,i] represents a discrete version of the continuous-time channel vector $h(k\Delta t,f_{i})$. Each dotted box corresponds to a MIMO channel model that includes $M_{r}M_{t}$ state-space SISO channel blocks, a spatial correlation matrix $\Phi_{M_{r}M_{t}}$, and a single carrier frequency. This setup creates $M_{f}$ frequency-selective channels in parallel. Additionally, the square matrix D of order $M_{f}$ in the block diagram is derived from the spectral correlation matrix $C_{f}$ as shown in Equation (Equation 28). This matrix D is used to adjust the spectral correlation properties among the MIMO channel blocks.

Mathematically, this MIMO-OFDM channel model can be represented by [25]
(37)$$\begin{array}{cc} x_{k+1} & =\Gamma x_{k}+\Psi w_{k}\\ h_{k} & =\Omega x_{k} \end{array}$$
where ${\left[h{\left[k,0\right]}^{T}\;h{\left[k,1\right]}^{T}\;\cdots \;h{\left[k,M_{f}-1\right]}^{T}\right]}^{T}\in C^{M_{f}M_{r}M_{t}}$ is denoted by $h_{k}$, $x_{k}\in C^{{\mathrm{pM}}_{f}M_{r}M_{t}}$, $w_{k}\in C^{M_{f}M_{r}M_{t}}$, Γ is a complex square matrix of order $pM_{f}M_{r}M_{t}$, Ψ is a $pM_{f}M_{r}M_{t}\times M_{f}M_{r}M_{t}$ complex matrix, Ω is a $M_{f}M_{r}M_{t}$ by $pM_{f}M_{r}M_{t}$ real matrix. The matrices Γ, Ψ and Ω are given by
(38)$$\Gamma =I_{M_{f}}⊗\Gamma^{mimo},\;\Psi =D⊗\Psi^{mimo},\;\Omega =I_{M_{f}}⊗\Omega^{mimo}$$
where *p* is the order of the AR model, $\Gamma^{mimo}$, $\Psi^{mimo}$ and $\Omega^{mimo}$ are given by Equation (Equation 25), and D is defined as a lower triangular matrix that satisfies the following:(39)$$DD^{H}=C_{f}$$

Similarly, the Cholesky decomposition method can be used to determine all elements of the lower triangular matrix D. However, when working with two carrier frequencies, a closed-form solution can be derived through straightforward algebraic operations, as outlined in the following:(40)$$\begin{array}{c} D=\left[\begin{array}{cc} 1 & 0\\ r_{f}^{*}\left[m_{f}\right] & \sqrt{1-|r_{f}\left[m_{f}\right]{|}^{2}} \end{array}\right] \end{array}$$

Thus, an AR(p)-based state-space MIMO-OFDM channel model is developed using this method. It is important to highlight that this approach is specifically designed for a single scattering cluster. The following section will describe the methodology for constructing a multi-cluster MIMO-OFDM channel model.

#### 2.3.6. Multi-Cluster MIMO-OFDM Channel Model

By connecting multiple MIMO-OFDM channel model blocks in parallel, a multi-cluster MIMO-OFDM channel model is assembled, as illustrated in Figure 9. The number of blocks required depends on the value of *K*.

The configuration shown in Figure 9 can be expressed mathematically as follows [25]:(41)$$\begin{array}{cc} x_{k+1} & =\Gamma x_{k}+\Psi w_{k}\\ h_{k} & =\Omega x_{k} \end{array}$$
where Γ, Ψ, and Ω are given below:(42)$$\begin{array}{cc} & \Gamma =\left[\begin{array}{cccc} \Gamma_{1} & 0 & \cdots & 0\\ 0 & \Gamma_{2} & \cdots & 0\\ \vdots & \vdots & \ddots & \vdots \\ 0 & 0 & \cdots & \Gamma_{K} \end{array}\right],\;\;\;\Psi =\left[\begin{array}{cccc} \Psi_{1} & 0 & \cdots & 0\\ 0 & \Psi_{2} & \cdots & 0\\ \vdots & \vdots & \ddots & \vdots \\ 0 & 0 & \cdots & \Psi_{K} \end{array}\right],\\ \; & \;\Omega =\left[\begin{array}{cccc} \Omega_{1} & 0 & \cdots & 0\\ 0 & \Omega_{2} & \cdots & 0\\ \vdots & \vdots & \ddots & \vdots \\ 0 & 0 & \cdots & \Omega_{K} \end{array}\right] \end{array}$$
where $\Gamma_{i}$, $\Psi_{i}$, and $\Omega_{i}$ are defined in Equation (Equation 38). These matrices are derived either from the AR(3)-based MIMO-OFDM channel model or the AR(1)-based MIMO-OFDM channel model.

## 3. Simulations

In this section, we will select a 2×2 MIMO channel and a pair of carrier frequencies from the MIMO-OFDM channel model presented in Section 2. Through simulations, we will evaluate the performance of the selected MIMO-OFDM channel model within a multi-cluster environment.

Assume K=5, indicating five scattering clusters in the radio wave propagation environment, as illustrated in Figure 5. Among these, two clusters are Cauchy–Rayleigh clusters (clusters No. 3 and No. 4), while the remaining three are Rayleigh clusters (clusters No. 1, No. 2, and No. 5).

In Figure 5, let the origin be at the BS, with the BS positioned at (0, 0) and the MS at (3000, 0) along the x-axis. The Cartesian coordinate system is thus defined, with cluster centers located at $O_{1}$ (800, 700), $O_{2}$ (2400, 450), $O_{3}$ (1500, −350), $O_{4}$ (900, −1300), and O5 (2700, −900). Here, Oi denotes the center of cluster No. i. Table 1 provides the data used to determine the parameters for generating each cluster, with distances measured in meters.

For all scattering clusters, let $\beta_{90\%}=2{}^{\circ}$, the equation $P_{\beta }^{r}=\mathrm{erf}\left(\beta_{y\%}/\sqrt{2}\sigma_{t}\right)$ derived from the Rayleigh scattering clusters, the corresponding values are $\eta_{t}=0.0055$ and $\sigma_{t}=0.0212$ [25]. Consequently, the dispersion parameters of the clusters ζ and σ can be computed using the formulas provided in Equations (Equation 10) and (Equation 15), respectively. Table 2 presents the parameters necessary for plotting the angular delay spectrum for these five clusters, with average time delay measured in microseconds.

Figure 10a,b illustrate the characteristics of these five clusters. Specifically, Figure 10a presents the angular-delay spectrum of these clusters, showing how signal components are spread in both angle and delay. Meanwhile, Figure 10b depicts the corresponding power azimuth spectrum, highlighting the distribution of signal power over various azimuth angles for these clusters.

Figure 11a shows the power delay profile of the clusters. The histogram, representing this profile, is computed from $5\times 10^{4}$ scatterers. Similarly, Figure 11b depicts the power of the AOA for these clusters, where the histogram is also calculated based on $5\times 10^{4}$ scatterers.

In the simulation, two distant Cauchy–Rayleigh clusters and three distant Rayleigh clusters are assumed. Table 3 lists all the parameters required to construct a five-cluster MIMO-OFDM channel model.

Additionally, let γ=−50°, $d_{t}=d_{r}=m_{f}=1$ and Δf=1 MHz, then this five-cluster MIMO-OFDM channel model can be represented by Equation (Equation 41), where D is defined in Equation (Equation 40) with elements derived either from Equation (Equation 33) or Equation (Equation 36). The matrices $\Gamma_{1},\Gamma_{2},\Gamma_{5}\in \Gamma$ are all 24×24 matrices of the AR(3) model, while $\Gamma_{3},\Gamma_{4}\in A$ are both 8×8 matrices of the AR(1) model. The same applies to the matrices Ψ and Ω. Finally, the output corresponding to each cluster is a vector containing eight channels.

The following expressions
(43)$$\begin{array}{cc} {\tilde{C}}_{h}^{c}\left(\Delta t,d_{t},d_{r},d_{f}\right)= & \;e^{-2\pi \eta d_{f}}e^{-j2\pi d_{f}\bar{\tau }}e^{-2\pi \eta_{t}d_{t}\left|\sin\left(\beta_{0}\right)\right|}e^{j2\pi d_{t}\cos\left(\beta_{0}\right)}\\ & \;\times e^{-2\pi \eta_{r}\left|d_{r}\sin\left(\alpha_{0}\right)+f_{D}\Delta t\sin\left(\alpha_{0}-\gamma \right)\right|}e^{j2\pi (d_{r}\cos\left(\alpha_{0}\right)+f_{D}\Delta t\cos\left(\alpha_{0}-\gamma \right))}\\ {\tilde{C}}_{h}^{r}\left(\Delta t,d_{t},d_{r},d_{f}\right)= & \;e^{-2\pi^{2}\sigma_{0}^{2}d_{f}^{2}}e^{-j2\pi \bar{\tau }d_{f}}e^{-2\pi^{2}\sigma_{t}^{2}d_{t}^{2}\sin^{2}\left(\beta_{0}\right)}e^{j2\pi d_{t}\cos\left(\beta_{0}\right)}\\ & \;\times e^{-2\pi^{2}\sigma_{r}^{2}d_{r}^{2}\sin^{2}\left(\alpha_{0}\right)}e^{j2\pi d_{r}\cos\left(\alpha_{0}\right)}e^{-4\pi^{2}\sigma_{r}^{2}d_{r}\sin\left(\alpha_{0}\right)\sin\left(\alpha_{0}-\gamma \right)f_{D}\Delta t}\\ \; & \;\times e^{-2\pi^{2}\sigma_{r}^{2}\sin^{2}\left(\alpha_{0}-\gamma \right)f_{D}^{2}\Delta t^{2}}e^{j2\pi \cos\left(\alpha_{0}-\gamma \right)f_{D}\Delta t} \end{array}$$
represent the STSCFs of the channel. The channel STCFs, ${\tilde{C}}_{h}^{c}\left(\Delta t,d_{t},d_{r},0\right)$ and ${\tilde{C}}_{h}^{r}\left(\Delta t,d_{t},d_{r},0\right)$, are given by [25], which describe the contributions of a single Cauchy–Rayleigh cluster and a Rayleigh cluster to the channel, respectively.

In simulation, using Equation (43) and combining the contributions of all the individual clusters, we obtain the channel correlation characteristics described by this five-cluster environment, referred to as the analytical model.

Correspondingly, the following are the STSCFs of the channel in the phase-shift expressions for a single Cauchy–Rayleigh cluster and a Rayleigh cluster:(44)$$\begin{array}{cc} {\tilde{C}}_{h}^{c_{\kappa }}\left(\Delta t_{\kappa },d_{t},d_{r},d_{f}\right)= & \;{\tilde{R}}_{h}^{c_{\kappa }}\left(\Delta t_{\kappa }\right)r_{m,n}^{c,\mathrm{BS}}\left(d_{t}\right)r_{i,j}^{\mathrm{MS}}\left(d_{r}\right)r_{f}^{c}\left(d_{f}\right)\\ {\tilde{C}}_{h}^{r_{\kappa }}\left(\Delta t_{\kappa },d_{t},d_{r},d_{f}\right)= & \;{\tilde{R}}_{h}^{r_{\kappa }}\left(\Delta t_{\kappa }\right)r_{m,n}^{r,\mathrm{BS}}\left(d_{t}\right)r_{i,j}^{\mathrm{MS}}\left(d_{r}\right)r_{f}^{r}\left(d_{f}\right) \end{array}$$
where ${\tilde{C}}_{h}^{c_{\kappa }}\left(\Delta t_{\kappa },d_{t},d_{r},0\right)$ and ${\tilde{C}}_{h}^{r_{\kappa }}\left(\Delta t_{\kappa },d_{t},d_{r},0\right)$ are from Equations (Equation 11) and (Equation 16), respectively.

In simulation, using Equation (44) and combining the contributions from each AR(p)-based state-space MIMO-OFDM channel model, we can construct state-space MIMO-OFDM channels with multiple clusters.

Figure 12, Figure 13, Figure 14, Figure 15 and Figure 16 display the real parts of the STCFs and STSCFs for individual clusters along with their corresponding state-space model estimations. For each cluster, 400,000 channel samples were generated to estimate the correlation properties. These figures demonstrate that the estimation closely matches the channel correlation function within the distance of interest. Therefore, an AR(p)-based state-space MIMO-OFDM channel model can effectively be used to describe the channel correlation properties for each cluster.

Figure 17a illustrates a segment of the real part of the SISO channel sequence generated by this five-cluster state-space MIMO-OFDM channel model. This sequence is utilized to estimate the SISO channel correlation function shown in Figure 17b.

Figure 18a,b display the real part of the SIMO and MISO channel correlation functions for the five scattering clusters, respectively. The estimations align closely with the analytical models.

Figure 19a,b depict the real part of the MIMO and MIMO-OFDM channel correlation functions for the five scattering clusters. The estimated curves follow the analytical models very well.

Thus, the AR(p)-based state-space MIMO-OFDM channel model can effectively be employed to describe a multi-cluster radio wave propagation environment.

It is important to note that the phase-shift method is used to construct the state-space MIMO-OFDM channel model for each scattering cluster. Consequently, each state-space channel model has its own initial value of movement. When sampling the moving distances of interest, a phase difference may occur between the channels generated by these state-space channel models, leading to errors when combining these channels. However, if a smaller sampling period is chosen, this error becomes negligible.

## 4. Conclusions

This paper presents a multi-cluster approach to radio sensor array channel modeling and simulations, which is crucial for integrating the contributions from multiple clusters, enabling the use of an AR(p) model to characterize channel dynamics. Subsequently, an AR(p)-based state-space model is developed for the radio channel, facilitating the creation of a multi-cluster MIMO-OFDM channel model.

The key benefit of this proposed phase-shift method is its ability to modify the variables to prevent decomposition errors that arise from excessive approximations, thereby avoiding errors in the radio channel model derived from channel correlation functions.

By incorporating antenna spacing into the motion component, this approach allows us to deal with channel correlation properties in the transformed system and then map them back to the original system using an inverse linear transformation. Simulations confirm that this phase-shifting technique effectively resolves the decomposition problem and validates that classifying the physical propagation environment of radio waves into scattering clusters is a suitable approach for radio channel modeling. Additionally, simulations demonstrate that the spatial–temporal correlation characteristics of the channel are accurately captured in a multi-cluster environment.

Furthermore, the paper addresses the limitations associated with analyzing CSOS using small angle ranges. It explores the extension of the critical angle from [−π,π] to (−∞,∞), without compromising the essential characteristics of the CSOS.

The modular construction of radio channels results in a state-space-based MIMO-OFDM channel model. Each distant scattering cluster contributing to an antenna at a mobile receiver is modeled using AR(1) or AR(3) state-space SISO channel blocks. The advantage of the state-space representation is that it enables the construction of a MIMO-OFDM channel model from multiple SISO channel blocks. Additionally, a correlated innovation process is employed to adjust the spatial correlation within each MIMO block and the spectral correlation between MIMO blocks. This framework can be extended to multi-cluster scenarios using the same methodology.

Therefore, this approach is both valid and applicable to any multi-cluster MIMO-OFDM channel model and can also be used for constructing massive MIMO channels. It effectively enables the simulation of physical radio wave propagation environments.

Although the current work focuses on multi-cluster MIMO-OFDM channel modeling, there are several avenues for further research and extensions:Massive MIMO and Beyond 5G Applications:The proposed model is applicable to massive MIMO configurations beyond 5G and 6G networks, where the number of antennas and channel complexity increase significantly. Future work could explore the scalability of the phase-shift approach in these larger and more complex systems.Hybrid Beamforming Integration:With the rise of hybrid beamforming in MIMO systems, this model can be used to analyze and optimize beamforming strategies in a multi-cluster environment. This extension is particularly important in mmWave and THz communications.Machine Learning Integration:By integrating machine learning techniques, this model can be used to dynamically optimize parameters or predict channel characteristics based on historical data, which will make this radio channel model more adaptable and intelligent, especially in environments with complex or rapidly changing conditions.

## Figures and Tables

**Figure 1 sensors-24-06037-f001:**
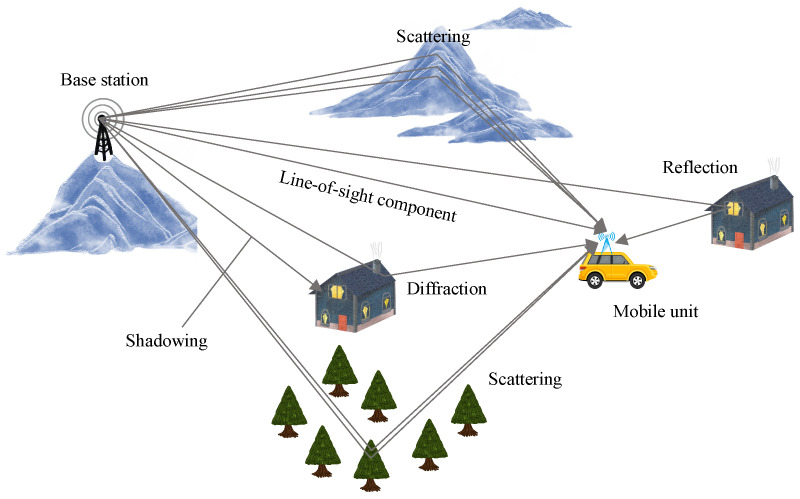
A typical mobile radio scenario for multi-path propagation in a terrestrial radio propagation environment.

**Figure 2 sensors-24-06037-f002:**
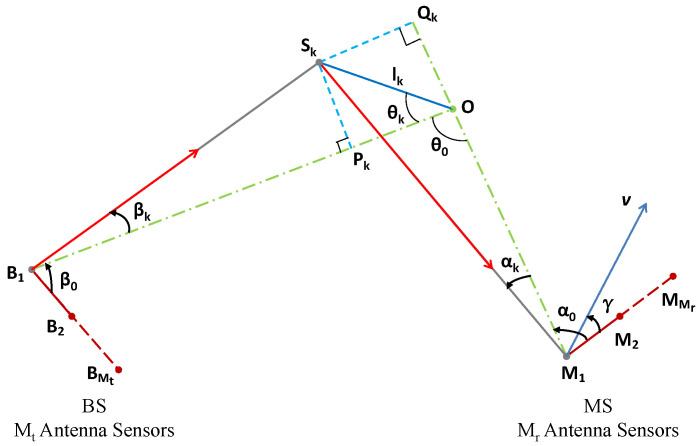
A distant cluster with an $M_{r}\times M_{t}$ MIMO antenna array, $l_{k}$ is the distance between the scattering object $S_{k}$ and the cluster center O.

**Figure 3 sensors-24-06037-f003:**
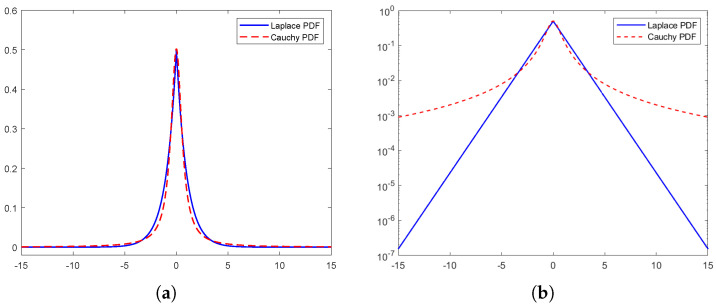
Laplace PDF and Cauchy PDF, special case of parametrization. (**a**) Laplace PDF $g_{x}\left(x\right)\;=\;\frac{1}{2}e^{-|x|}$. (**b**) Cauchy PDF $f_{x}\left(x\right)=\frac{1}{\pi }\frac{\eta }{\eta^{2}+x^{2}}$, where η=0.634.

**Figure 4 sensors-24-06037-f004:**
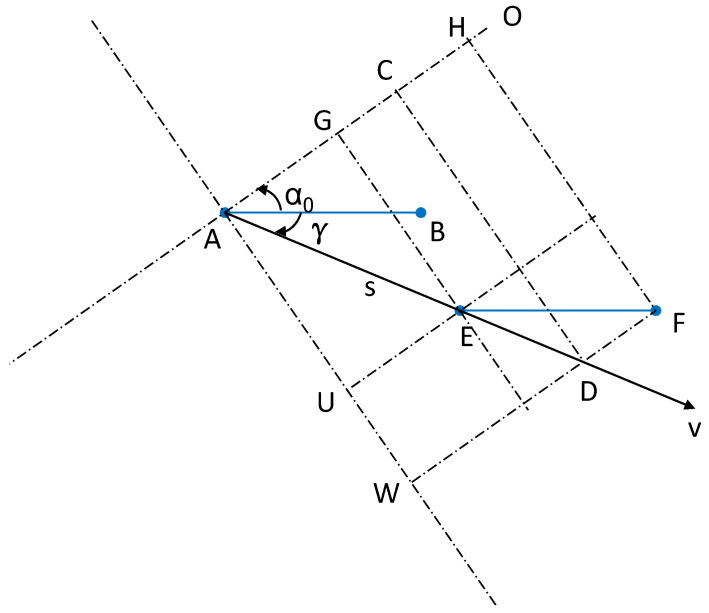
The motion vector is decomposed into a phase change on OA and a damping change on AW, where AB and EF denote the antenna arrays, and A, B, E, and F are antenna sensors.

**Figure 5 sensors-24-06037-f005:**
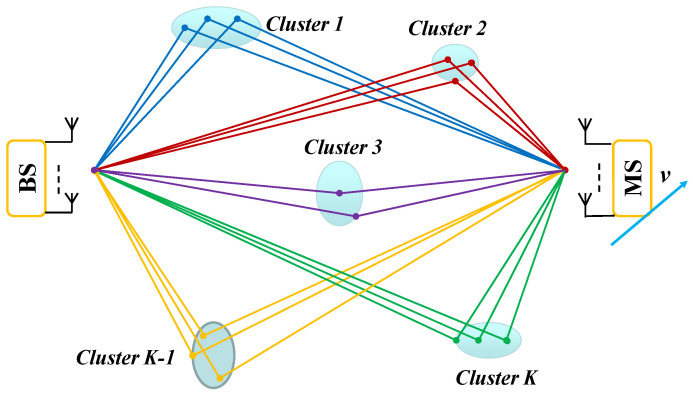
K distant scattering clusters, cluster no. 1 to cluster no. K, in a radio wave propagation environment, each of which is broken down into many resolvable multi-path components.

**Figure 6 sensors-24-06037-f006:**
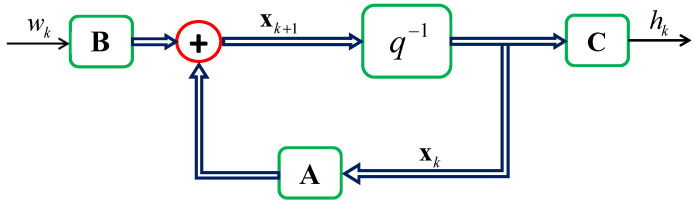
Block diagram of the AR(p)-based state-space SISO channel model.

**Figure 7 sensors-24-06037-f007:**
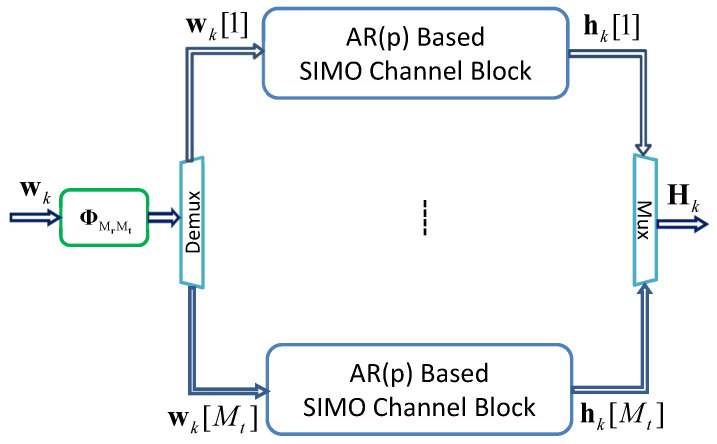
Block diagram of the AR(p)-based state space MIMO channel model, where $h_{k}\left[m\right]$, *m* denotes channel index.

**Figure 8 sensors-24-06037-f008:**
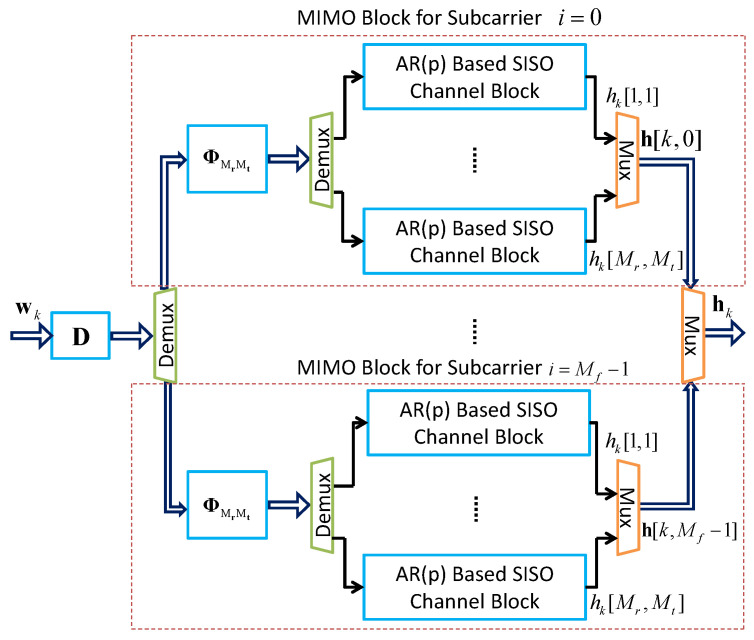
Block diagram of the MIMO-OFDM channel model, where $h_{k}\left[m,n\right]$, *m* and *n* denote channel index.

**Figure 9 sensors-24-06037-f009:**
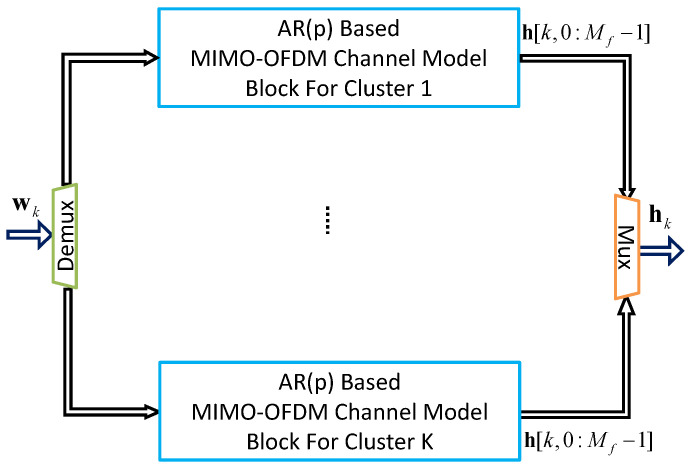
Block diagram of a *K*-cluster MIMO-OFDM channel model, where the input noise vector $w_{k}\in C^{{\mathrm{KM}}_{f}M_{r}M_{t}}$, the output channel vector $h[k,0:M_{f}-1]$ means that there are $M_{f}$ sub-carriers from $m_{f}=0$ to $m_{f}=M_{f}-1$.

**Figure 10 sensors-24-06037-f010:**
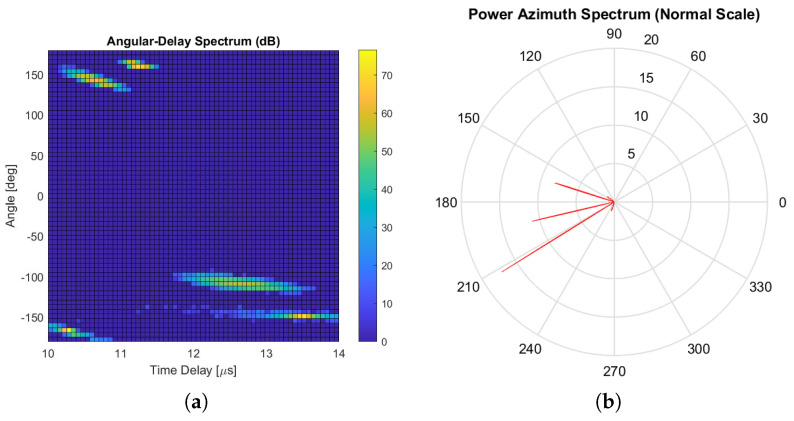
Angular-delay spectrum and power azimuth spectrum of the clusters. (**a**) Angular-delay spectrum of the clusters. (**b**) Power azimuth spectrum of the clusters.

**Figure 11 sensors-24-06037-f011:**
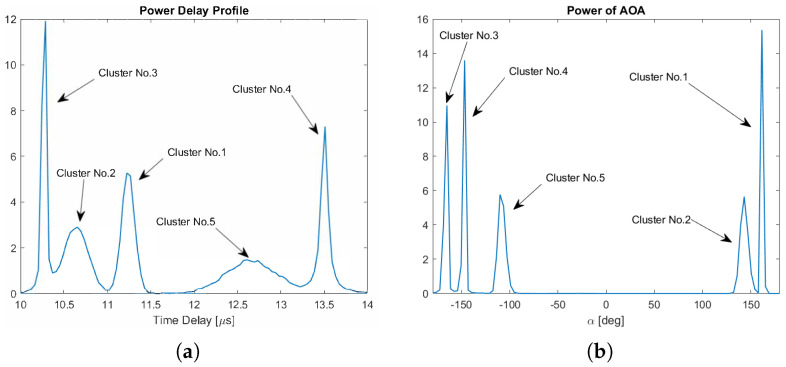
The power of AOA and power delay profile of the clusters. (**a**) The PDP of the clusters, where ${\bar{\tau }}_{1}=11.24\;\mu s$, ${\bar{\tau }}_{2}=10.64\;\mu s$, ${\bar{\tau }}_{3}=10.27\;\mu s$, ${\bar{\tau }}_{4}=13.5\;\mu s$, ${\bar{\tau }}_{5}=12.65\;\mu s$. (**b**) The power of AOA of the clusters, where $\alpha_{0_{1}}=162.35$°, $\alpha_{0_{2}}=143.13$°, $\alpha_{0_{3}}=-166.87$°, $\alpha_{0_{4}}=-148.24$°, $\alpha_{0_{5}}=-108.43$°.

**Figure 12 sensors-24-06037-f012:**
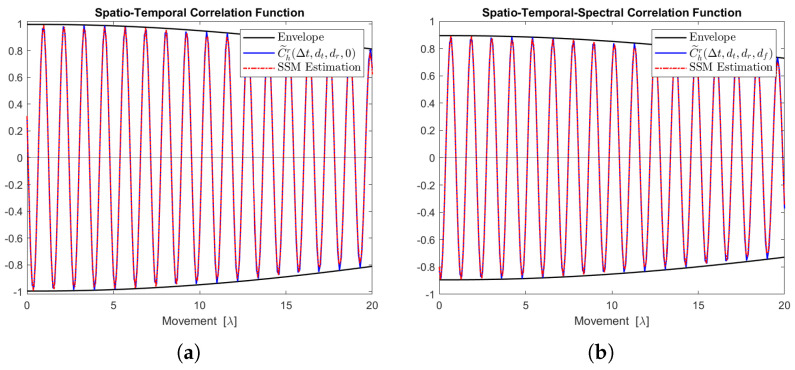
Channel correlation estimation of cluster No. 1, characterized as a Rayleigh cluster. The sequence is generated from the state-space MIMO-OFDM channel model. (**a**) The real part of the STCF, where $d_{t}=d_{r}=1$. (**b**) The real part of the STSCF, where $d_{t}=d_{r}=m_{f}=1$, Δf=1 MHz.

**Figure 13 sensors-24-06037-f013:**
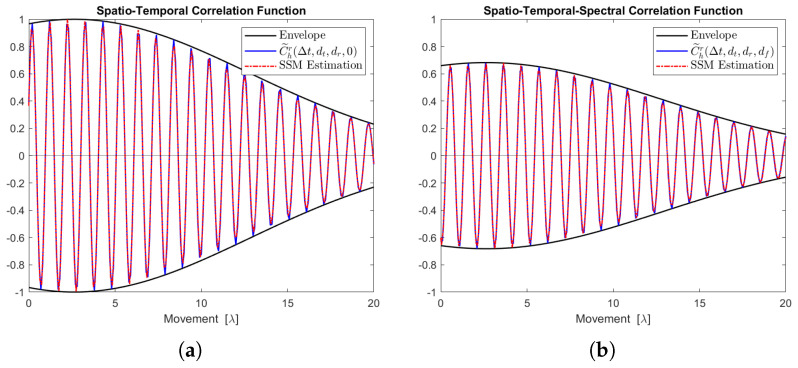
Channel correlation estimation of cluster No. 2, characterized as a Rayleigh cluster. The sequence is generated from the state-space MIMO-OFDM channel model. (**a**) The real part of the STCF, where $d_{t}=d_{r}=1$. (**b**) The real part of the STSCF, where $d_{t}=d_{r}=m_{f}=1$, Δf=1 MHz.

**Figure 14 sensors-24-06037-f014:**
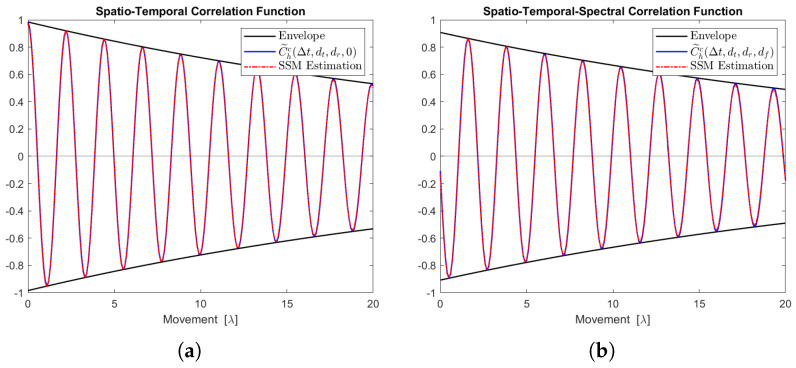
Channel correlation estimation of cluster No. 3, characterized as a Cauchy–Rayleigh cluster. The sequence is generated from the state-space MIMO-OFDM channel model. (**a**) The real part of the STCF, where $d_{t}=d_{r}=1$. (**b**) The real part of the STSCF, where $d_{t}=d_{r}=m_{f}=1$, Δf=1 MHz.

**Figure 15 sensors-24-06037-f015:**
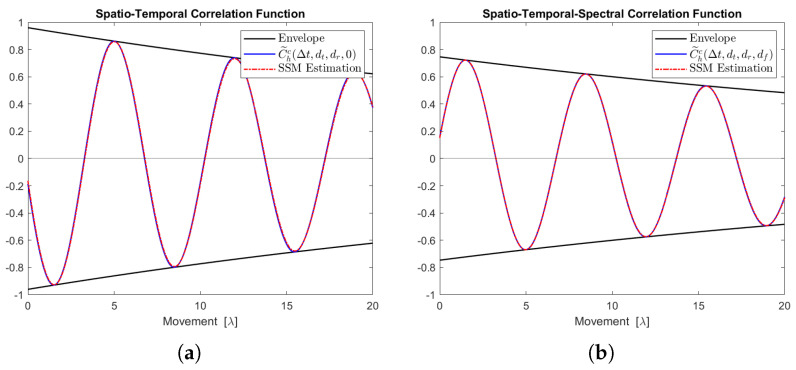
Channel correlation estimation of cluster No. 4, characterized as a Cauchy–Rayleigh cluster. The sequence is generated from the state-space MIMO-OFDM channel model. (**a**) The real part of the STCF, where $d_{t}=d_{r}=1$. (**b**) The real part of the STSCF, where $d_{t}=d_{r}=m_{f}=1$, Δf=1 MHz.

**Figure 16 sensors-24-06037-f016:**
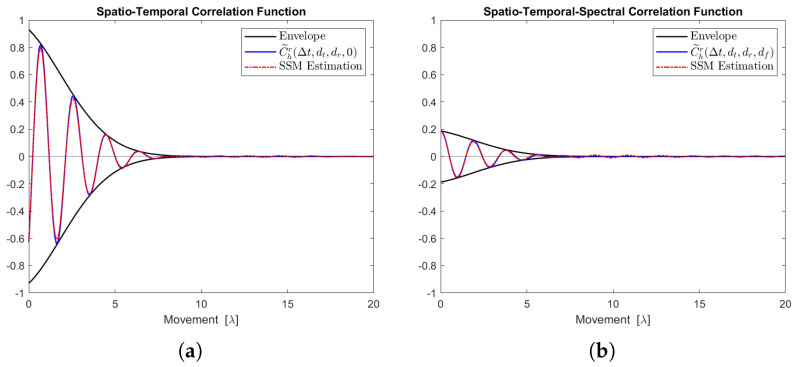
Channel correlation estimation of cluster No. 5, characterized as a Rayleigh cluster. The sequence is generated from the state-space MIMO-OFDM channel model. (**a**) The real part of the STCF, where $d_{t}=d_{r}=1$. (**b**) The real part of the STSCF, where $d_{t}=d_{r}=m_{f}=1$, Δf=1 MHz.

**Figure 17 sensors-24-06037-f017:**
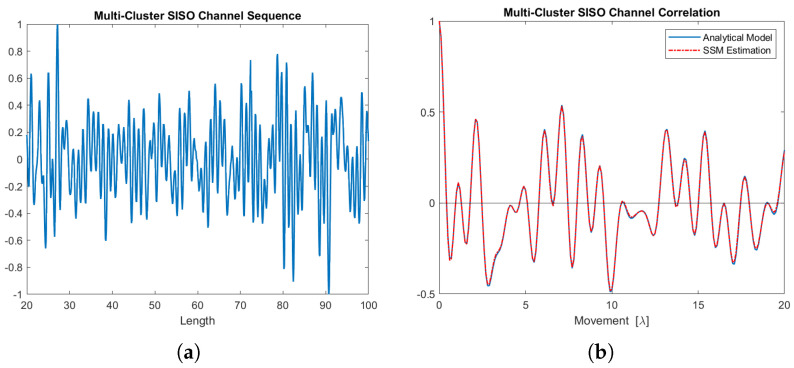
SISO channel sequence with its corresponding correlation function. (**a**) The SISO channel sequence (real part) generated by this 5-cluster state-space MIMO-OFDM channel model. (**b**) The SISO channel correlation function (real part), estimation is from this 5-cluster state-space MIMO-OFDM channel model.

**Figure 18 sensors-24-06037-f018:**
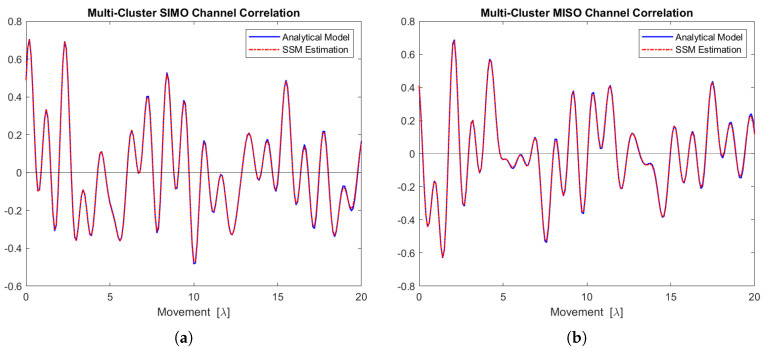
Channel correlation estimation based on the sequence generated from this 5-cluster state-space MIMO-OFDM channel model. (**a**) The SIMO channel correlation function (real part). (**b**) The MISO channel correlation function (real part).

**Figure 19 sensors-24-06037-f019:**
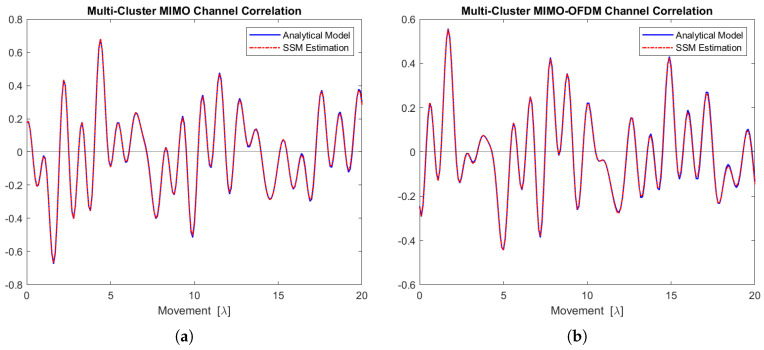
Channel correlation estimation based on the sequence generated from this 5-cluster state-space MIMO-OFDM channel model. (**a**) The MIMO channel correlation function (real part). (**b**) The MIMO-OFDM channel correlation function (real part).

**Table 1 sensors-24-06037-t001:** The parameters used to generate distant scattering clusters.

Cluster No. 1	Cluster No. 2	Cluster No. 3	Cluster No. 4	Cluster No. 5
Rayleigh	Rayleigh	Cau.–Rayleigh	Cau.–Rayleigh	Rayleigh
$d_{{\mathrm{OB}}_{1}}=1063$	$d_{{\mathrm{OB}}_{1}}=2442$	$d_{{\mathrm{OB}}_{1}}=1540$	$d_{{\mathrm{OB}}_{1}}=1581$	$d_{{\mathrm{OB}}_{1}}=2846$
$d_{{\mathrm{OM}}_{1}}=2309$	$d_{{\mathrm{OM}}_{1}}=750$	$d_{{\mathrm{OM}}_{1}}=1540$	$d_{{\mathrm{OM}}_{1}}=2470$	$d_{{\mathrm{OM}}_{1}}=949$
$\cos(\theta_{0})=121.16{}^{\circ}$	$\cos(\theta_{0})=132.51{}^{\circ}$	$\cos(\theta_{0})=$ 153.73°	$\cos(\theta_{0})=$ 92.94°	$\cos(\theta_{0})=$ 90°
$\beta_{0}=$ 41.19°	$\beta_{0}=$ 10.26°	$\beta_{0}=$ −13.13°	$\beta_{0}=$ −55.31°	$\beta_{0}=$ −18.84°
$\alpha_{0}=$ 162.35°	$\alpha_{0}=$ 143.13°	$\alpha_{0}=$ −166.87°	$\alpha_{0}=$ −148.24°	$\alpha_{0}=$ −108.43°

**Table 2 sensors-24-06037-t002:** The parameters used to draw the graphs.

Cluster No. 1	Cluster No. 2	Cluster No. 3	Cluster No. 4	Cluster No. 5
Rayleigh	Rayleigh	Cau.–Rayleigh	Cau.–Rayleigh	Rayleigh
σ=22.561	σ=51.825	ζ=8.516	ζ=8.742	σ=60.404
$\sigma_{r}=0.01$	$\sigma_{r}=0.069$	$\eta_{r}=0.006$	$\eta_{r}=0.004$	$\sigma_{r}=0.064$
$\bar{\tau }=11.24$	$\bar{\tau }=10.64$	$\bar{\tau }=10.27$	$\bar{\tau }=13.5$	$\bar{\tau }=12.65$
$\sigma_{0}=$ 7.4 × 10^−8^	$\sigma_{0}=$ 1.4 × 10^−7^	η= 1.3 × 10^−8^	η= 4 × 10^−8^	$\sigma_{0}=$ 2.9 × 10^−7^

**Table 3 sensors-24-06037-t003:** The parameters used to build the five-cluster MIMO-OFDM channel model.

Cluster No. 1	Cluster No. 2	Cluster No. 3	Cluster No. 4	Cluster No. 5
Rayleigh	Rayleigh	Cau.–Rayleigh	Cau.–Rayleigh	Rayleigh
$\sigma_{t}=0.021$	$\sigma_{t}=0.021$	$\eta_{t}=0.006$	$\eta_{t}=0.006$	$\sigma_{t}=0.021$
$\sigma_{r}=0.01$	$\sigma_{r}=0.069$	$\eta_{r}=0.006$	$\eta_{r}=0.004$	$\sigma_{r}=0.064$
$\bar{\tau }=11.24$	$\bar{\tau }=10.64$	$\bar{\tau }=10.27$	$\bar{\tau }=13.5$	$\bar{\tau }=12.65$
$\sigma_{0}=$ 7.4 × 10^−8^	$\sigma_{0}=$ 1.4 × 10^−7^	η= 1.3 × 10^−8^	η= 4 × 10^−8^	$\sigma_{0}=$ 2.9 × 10^−7^
$\beta_{0}=$ 41.19°	$\beta_{0}=$ 10.62°	$\beta_{0}=$ −13.13°	$\beta_{0}=$ −55.31°	$\beta_{0}=$ −18.44°
$\alpha_{0}=$ 162.35°	$\alpha_{0}=$ 143.13°	$\alpha_{0}=$ −166.87°	$\alpha_{0}=$ −148.24°	$\alpha_{0}=$ −108.43°

## Data Availability

Data are contained within the article.

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
