# Peer review of "Multi-Cluster Approaches to Radio Sensor Array Channel Modeling and Simulation"

_sensors, 2024, doi:10.3390/s24186037_

Round 1

Reviewer 1 Report

Comments and Suggestions for Authors

This paper investigates channel modeling for MIMO systems using second-order statistics. The topic is timely and suitable for publication in the *Sensors* journal. However, several weaknesses need to be addressed:

- The authors do not specify a frequency range, making it difficult to assess the applicability of their model across all radio frequency ranges. It is important to note that mmWave and THz communications exhibit behaviors that differ significantly from sub-6GHz channel modeling and estimation. The reviewer encourages the authors to specify the frequency range relevant to their study.

- Secondly, the geometry-based approach may not be suitable for all types of environments, as it is typically designed for specific areas.

- Third, channel correlation should be considered for a more general case.

- In the simulation, a 2x2 system is considered, which may not be appropriate given that modern base stations typically employ a much larger number of antennas, often in the tens or hundreds, as seen in massive MIMO systems.

- The authors should conduct practical experiments to strengthen their contributions, particularly since channel modeling can be considered outdated when the carrier frequency is sub-6GHz.

- Clarify how the proposed method is superior to the state-of-the-art. The authors should carefully justify the proposed method.

- In the conclusion section, offer more discussions about possible extensions of the proposed approach.

- Keywords should be presented in alphabetical order for better organization.

- Carefully proofread the manuscript to eliminate any remaining typos or errors before submission.

- If possible, add DOIs for all references and strictly adhere to the conference template.

Comments on the Quality of English Language

It is okay

Reviewer 2 Report

Comments and Suggestions for Authors

This paper investigated the physical propagation environment of radio waves. By describing the radio wave propagation environment in terms of scattering clusters, a multi-cluster approach was proposed. Simulations were performed to validate the proposed method. The work is solid, and the paper is well-written. I have a few comments and questions:

1. The text in some figures (for example, Figures 12-19) is too small.

2. Please explain what the analysis model in Figure 17-19 refers to.

3. How do the different types of clusters correspond to the real-world targets? Please comment on this.

4. Is the interaction of different clusters necessary to be considered? Please comment on this.

Round 2

Reviewer 1 Report

Comments and Suggestions for Authors

I have no more comments.

Comments on the Quality of English Language

It is fine.